# Delayed Drug Hypersensitivity Reactions: Molecular Recognition, Genetic Susceptibility, and Immune Mediators

**DOI:** 10.3390/biomedicines11010177

**Published:** 2023-01-10

**Authors:** Mu-Tzu Chu, Wan-Chun Chang, Shih-Cheng Pao, Shuen-Iu Hung

**Affiliations:** 1Cancer Vaccine & Immune Cell Therapy Core Lab, Department of Medical Research, Chang Gung Memorial Hospital, Linkou Branch, Taoyuan 333, Taiwan; 2Division of Translational Therapeutics, Department of Paediatrics, Faculty of Medicine, University of British Columbia, Vancouver, BC V6T 1Z4, Canada; 3Institute of Pharmacology, School of Medicine, National Yang Ming Chiao Tung University, Taipei 112, Taiwan; 4Drug Hypersensitivity Clinical and Research Center, Department of Dermatology, Chang Gung Memorial Hospital, Linkou 333, Taiwan

**Keywords:** drug hypersensitivity, T lymphocytes, HLA, Stevens–Johnson syndrome, drug reaction with eosinophilia and systemic symptoms, severe cutaneous adverse reactions

## Abstract

Drug hypersensitivity reactions are classified into immediate and delayed types, according to the onset time. In contrast to the immediate type, delayed drug hypersensitivity mainly involves T lymphocyte recognition of the drug antigens and cell activation. The clinical presentations of such hypersensitivity are various and range from mild reactions (e.g., maculopapular exanthema (MPE) and fixed drug eruption (FDE)), to drug-induced liver injury (DILI) and severe cutaneous adverse reactions (SCARs) (e.g., Stevens–Johnson syndrome (SJS), toxic epidermal necrolysis (TEN), drug reaction with eosinophilia and systemic symptoms (DRESS), and acute generalized exanthematous pustulosis (AGEP)). The common culprits of delayed drug hypersensitivity include anti-epileptics, antibiotics, anti-gout agents, anti-viral drugs, etc. Delayed drug hypersensitivity is proposed to be initiated by different models of molecular recognition, composed of drug/metabolite antigen and endogenous peptide, HLA presentation, and T cell receptor (TCR) interaction. Increasing the genetic variants of HLA loci and drug metabolic enzymes has been identified to be responsible for delayed drug hypersensitivity. Furthermore, preferential TCR clonotypes, and the activation of cytotoxic proteins/cytokines/chemokines, are also involved in the pathogenesis of delayed drug hypersensitivity. This review provides a summary of the current understanding of the molecular recognition, genetic susceptibility, and immune mediators of delayed drug hypersensitivity.

## 1. Introduction

Drug hypersensitivity reactions are initiated by exposure to the drug within the therapeutic range, and present in immune-mediated characteristics and symptoms [1,2]. Most of the reactions are unpredictable adverse drug reactions (ADRs) and affect more than 7% of the general population worldwide [3]. Drug hypersensitivity reactions involve specific antibodies or T cell receptors (TCR). According to its definition by the Nomenclature Review Committee of the World Allergy Organization, drug hypersensitivity accounts for 15% of all kinds of drug-related adverse reactions [1,3,4]. The International Consensus on Drug Allergies (ICON) classifies drug hypersensitivity reactions into immediate and non-immediate types, according to the onset time (Table 1). The immediate type usually refers to the symptoms appearing within 1–6 h after exposure to the suspected drugs. The common presentations of immediate-type drug hypersensitivity include angioedema, urticaria and anaphylaxis. The frequent culprits for immediate drug hypersensitivity include non-steroid anti-inflammatory drugs (NSAIDs), neuromuscular blocking agents (NMBA), aspirin, antibiotics, and vaccines (Table 1). By comparison, non-immediate type hypersensitivity frequently presents as a delayed-type drug reaction, in which the onset time is days to weeks after an initial exposure to the culprit drugs [1,5,6,7,8]. Delayed-type drug hypersensitivity frequently involves skin reactions, ranging from mild reactions (e.g., maculopapular exanthema (MPE), and fixed drug eruption (FDE)) to severe cutaneous adverse reactions (SCARs) (e.g., drug reaction with eosinophilia and systemic symptoms (DRESS) (also called DiHS, drug-induced hypersensitivity syndrome), Stevens–Johnson syndrome (SJS), toxic epidermal necrolysis (TEN), and acute generalized exanthematous pustulosis (AGEP)) (Table 1). In addition, there are other delayed drug reactions, such as drug-induced liver injury (DILI), or drug-specific reaction (e.g., abacavir hypersensitivity) [9,10,11]. The mortality rates for SJS and TEN are high, i.e., 5–10% for SJS, 30% for SJS/TEN overlap, and 30–50% for TEN [12,13,14,15]. The mortality rate is approximately 4% for AGEP and 10% for DRESS [10,16]. The common causative drugs for delayed-type drug hypersensitivity are anti-epileptic drugs (AED), antibiotics, allopurinol, anti-viral agents, and NSAIDs (Table 1).

## 2. Classification of Delayed-Type Drug Hypersensitivity

The delayed-type hypersensitivity reaction is also called a type IV reaction, classified by Gell and Coombs, and mainly involves T cell-antigen recognition, accompanied by the activation of other leukocytes [17]. According to the types of mostly involved T cells and their downstream mediators, the type IV reaction is sub-grouped as IVa, IVb, IVc, and IVd [18] (Figure 1). Type IVa, IVb, and IVd are mediated by T_H_1, T_H_2, and IL-8-producing T_H_ cells, respectively, and involve other inflammatory cells, such as macrophages, eosinophils, and neutrophils. The type IVa reaction mainly involves T_H_1 cell activation, which releases chemokines and cytokines, such as IFN-γ and TNF-β, to recruit and activate macrophages to produce inflammatory mediators, such as TNF-α [19]. Type IVa drug hypersensitivity is suggested to be associated with MPE and FDE. The type IVb reaction mainly involves T_H_2 cell activation, which releases IL4, IL13, IL-5, and eotaxin to activate eosinophils and mast cells. Type IVb drug hypersensitivity is suggested to mediate the pathogenesis of DRESS/DiHS and DILI [20,21,22,23]. The CXCL8/IL-8-producing Th cells mediate the type IVd reaction via producing CXCL8 and GM-CSF to activate neutrophils, which are predominately in drug-related AGEP. By comparison, type IVc drug hypersensitivity mainly involves cytotoxic T cells; here, cytotoxic T lymphocytes (CTL) directly kill target cells by releasing cytotoxic cytokines, including granulysin, granzymes, and perforin, and also by cellular contact through a Fas/FasL pathway [24]. Type IVc drug hypersensitivity is mainly involved in SJS/TEN (Figure 1).

## 3. Proposed Models of Molecular Recognition in Delayed Drug Hypersensitivity

Delayed drug hypersensitivity reactions are initiated by a T cell receptor (TCR) recognizing the drug/metabolite antigen(s). The drug/metabolite antigen may interact with endogenous peptides covalently or noncovalently. The antigen could be presented with a human leukocyte antigen (HLA) to TCR, in order to induce T cell-mediated hypersensitivity reactions. There are four hypotheses proposed for the molecular recognition of drugs by TCR in delayed-type drug hypersensitivity: (1) the hapten hypothesis, (2) the pharmacological interaction with immune receptors (p-i) concept, (3) the altered peptide repertoire model, and (4) altered TCR repertoire model (Figure 2).
(1)The hapten/pro-hapten hypothesis describes that the causative drugs or the reactive metabolites are too small, with a molecular weight of fewer than 1000 daltons, to be immunogenic and recognized by the immune receptors. The haptens become immunogenic by the covalent binding of drug/metabolite to the endogenous peptides or proteins to form a hapten–carrier complex. The antigenic complex could be recognized by an antibody, or be presented on the HLA molecule and then recognized by TCR, resulting in the induction of drug-specific cellular or humoral immune responses. This hypothesis has been valid in cases of penicillin-induced ADRs [25,26,27]. The major antigenic determinant of penicillin-induced hypersensitivity is penicilloyl polylysine. This structure is formed by the covalent bond of a β-lactam ring to lysine residues in proteins [27,28]. Regarding the delayed drug hypersensitivty, penicilloyl peptides were found to be recognized as T-cell antigenic determinants in the penicillin allergy [25].(2)The pharmacological interaction with the immune receptor (p-i) concept postulates that drugs may noncovalently interact with the HLA, TCR, or endogenous proteins (or peptides) [29]. Our previous studies showed that carbamazepine (CBZ), one of the aromatic antiepileptic drugs, directly interacts with HLA-B*15:02 protein. This interaction of CBZ presentation on HLA-B*15:02 does not involve intracellular antigen processing or drug metabolism [30]. In addition, we showed another similar example for the interaction between oxypurinol and HLA-B*58:01. Oxypurinol, a reactive metabolite of allopurinol, can directly and immediately activate specific T cells through HLA-B*58:01; this is without intracellular antigen processing [31]. We demonstrated the key residuals of oxypurinol recognition on the HLA-B*58:01 cleft [32].(3)The altered peptide repertoire model refers to the causative drugs occupying a position in the peptide-binding groove of the HLA protein, the alteration of the properties of the binding cleft, and the peptide specificity of HLA binding. This model has been suggested by studies on abacavir hypersensitivity [33,34]. Abacavir binds to the F-pocket of HLA-B*57:01 and changes the properties of conformation and structure in the antigen-binding cleft. The interaction between the drug and HLA causes the altered peptide repertoire, resulting in TCR recognition, T cell activation, and a drug hypersensitivity reaction. The altered peptide repertoire causes a polyclonal activation of T cells and systemic manifestations resembling an autoimmune response [33,34].(4)The altered TCR repertoire model proposes that culprit drugs directly interact with TCR, and not with the peptides nor the HLA molecules. The antigenic molecules bind to specific TCRs and cause conformational change. The antigen-bound TCRs can interact with HLA-endogenous peptide complexes and elicit immune reactions [35]. In this model, the TCR repertoire is altered upon interaction with the drug/metabolite antigen. The drug antigen-bound TCR is essential in this model to induce drug hypersensitivity reactions [35].


In addition to the above four hypotheses, viral infection has been proposed to contribute to HLA/drug/TCR interactions, and viral peptides may be involved in the process of drug presentation and immune recognition, leading to drug hypersensitivity [26].

## 4. Genetic Susceptibility of Delayed Drug Hypersensitivity

Different approaches have been applied to explore the genetic susceptibility of drug hypersensitivity. The genetic variants involved in controlling (1) the immune response, especially the HLA alleles; (2) the drug metabolism enzymes for drug oxidation, conjugation, hydrolysis, and acetylation; and (3) the drug transporters or receptors, have been proposed to be associated with delayed-type drug hypersensitivity. The genetic susceptibility of delayed drug hypersensitivity showed drug-specific, phenotype-specific, and ethnic variation (Table 2, Table 3, Table 4, Table 5 and Table 6).

### 4.1. Genetic Susceptibility of Antiepileptics-Induced Hypersensitivity Reactions

We first reported that HLA-B*15:02 is a genetic marker for CBZ-induced SJS/TEN in Han Chinese patient populations in Taiwan in 2004 [36] (Table 2). This association has been further validated in other Asian countries, including Hong Kong, Singapore, Vietnam, Thailand, Malaysia, and India [37,38,39,40,41,42,43] (Table 2). We carried out a prospective study and showed that the genetic screening of HLA-B*15:02 before CBZ administration prevented the occurrence of CBZ-induced SJS/TEN [44]. None of the 4877 recruited patients who received CBZ treatment with preemptive pharmacogenomic testing developed SJS/TEN [44]. Additionally, we found HLA-B*57:01 was associated with CBZ-induced SJS/TEN in Europeans [45] (Table 2). By comparison, HLA-A*31:01 is associated with CBZ-induced MPE and DRESS, which was first reported in Han Chinese in Taiwan in 2006 [46]; this was then validated in different populations, including Europeans and Japanese [45,46,47,48,49,50,51] (Table 2).

Aside from being a risk allele for CBZ-induced SJS/TEN, HLA-B*15:02 has also been associated with SCARs induced by other antiepileptics that have a similar aromatic structure to CBZ, such as oxcarbazepine [52,53], phenytoin [40,53,54,55], and lamotrigine [56] (Table 2). In addition, HLA-A*32:01 was reported to be associated with oxcarbazepine-induced MPE in the Eastern Han Chinese population [57]. HLA-B*13:01 and B*51:01 are suggested to be related to phenytoin-induced SCARs in different studies in Asians, including Han Chinese, Japanese, and Malaysian patient populations [55,58] (Table 2). In addition to HLA alleles, we found that the loss of function in the allele of cytochrome P450 2C9 (CYP2C9), CYP2C9*3, affecting drug metabolism, was responsible for phenytoin-induced SCARs in Taiwan [58]. The genetic association was validated in the patient populations from Thailand and Japan [58,59,60] (Table 2). For lamotrigine-induced SCARs, HLA-A*31:01 and HLA-B*38:01 were reported to be risk alleles in patients of Asian or European descent [49,61] (Table 2).
biomedicines-11-00177-t002_Table 2Table 2Genetic variants associated with antiepileptics-induced hypersensitivity reactions.Causative DrugsReactionsGenetic FactorsEthnicityOR(95% CI)*p*-ValueReferenceCarbamazepine (CBZ)SJS/TENHLA-B*15:02Han Chinese2504(126–49,522)3.13 × 10^−27^[36]Thai25.5 (2.68–242.61)0.0005[40]7.27 (2.04–25.97)4.46 × 10^−13^[43]Indian71.40 (3.0–1698)0.0014[42]Malaysian16.15 (4.57–62.4)7.87 × 10^−6^[41]Vietnamese33.78 (7.55–151.03)<0.0001[39]Singaporean27.20 (2.67–∞)0.004[38]HLA-B*57:01European9.0 (4.2–19.4)9.62 × 10^−7^[45]DRESSHLA-A*31:01Han Chinese23.0 (4.2–125)<0.001[47]6.86 (2.4–19.9)2.7 × 10^−3^[50]European49.9 (12.9–193.6)4.0 × 10^−8^[45]13.2 (8.4–20.8)<0.001[47] 12.41 (1.27–121.03)3.5 × 10^−8^[48]22.00 (1.03–1190.36)0.047[49]Japanese10.8 (5.9–19.6)3.64 × 10^−15^[51]MPE/DRESSHLA-A*31:01Han Chinese17.5 (4.6–66.5)0.0022[46]MPEHLA-B*15:02Thai7.27 (2.04–25.97)0.0022[43]Oxcarbazepine (OXC)SJS/TENHLA-B*15:02Han Chinese27.90 (7.84–99.23)1.12 × 10^−9^[52]80.7 (3.8–1714.4)8.4 × 10^−4^[53]MPEHLA-A*32:01Han Chinese15.877 (1.817–138.720)0.004[57]Phenytoin (PHT)SCARsCYP2C9*3Taiwanese14.00 (6.75–29.02)0.00001[58]Japanese8.88 (2.20–35.83)Malaysian5.60 (0.56–56.20)
Thai4.30 (1.41–13.09)<0.05[59]
Taiwanese, Japanese, Thai20.86 (9.03–48.20)1.22 × 10^−13^[60]HLA-B*15:02Asian (Han Chinese, Japanese, Malaysian)5.0 (2.0–13)0.025[58]SJS/TENHLA-B*15:02Han Chinese5.1 (1.8–15.1)0.0041[53]3.50 (1.10–11.18)0.045[55]Thai18.5 (1.82–188.40)0.005[40]Malaysian5.71 (1.41–23.10)0.016[54]Lamotrigine (LTG)SCARsHLA-A*31:01Korean11.43 (1.95–59.77)0.0037[61]HLA-B*38:01European147.00 (1.88–483)0.001[49]SJS/TENHLA-B*15:02Han Chinese4.98 (1.43–17.28)0.01[56]AEDs (CBZ, LTG, PHT, etc.)SCARsHLA-B*15:02Han Chinese17.6 (2.9–105.2)0.001[37]Abbreviations: SJS/TEN, Stevens–Johnson syndrome/toxic epidermal necrolysis; DRESS, drug reaction with eosinophilia and systemic symptoms; MPE, maculopapular exanthema; OR, odds ratio; SCARs, severe cutaneous adverse reactions.

### 4.2. Genetic Susceptibility of Allopurinol Hypersensitivity

Allopurinol, a xanthine oxidase inhibitor, is the first-line drug to treat hyperuricemia and gout. Allopurinol is also one of the common culprit drugs to induce drug hypersensitivity. We first found HLA-B*58:01 to be associated with allopurinol-induced SCARs in Han Chinese people in Taiwan in 2005 [62] (Table 3). This association was further replicated and validated among various ethnicities, including European [63,64], Thai [65,66], Japanese [67], Korean [68], and African American [69] (Table 3). HLA-B*58:01 genetic screening has been shown to be a promising strategy for preventing allopurinol SCARs [70]. We further found that renal dysfunction, and increased plasma levels of the metabolite of allopurinol, i.e., oxypurinol, deteriorate the severity of allopurinol hypersensitivity [71]. This might explain the higher mortality rate of allopurinol-induced SCARs in patients with chronic kidney disease, because of the delayed clearance of oxypurinol. Furthermore, allopurinol-induced liver injury (DILI) was found to be associated with HLA-A*34:02, HLA-B*53:01, and HLA-B*58:01 [66,69] (Table 3). Some genomics studies validated the attribution of HLA-B*58:01 and proposed that other genetic variants, out of the HLA region, might also contribute to the development of allopurinol hypersensitivity [67] (Table 3).
biomedicines-11-00177-t003_Table 3Table 3Genetic variants associated with allopurinol-induced hypersensitivity reactions.ReactionsGenetic FactorsEthnicityOR(95% CI)*p*-ValueReferenceSJS/TENHLA-B*58:01Han Chinese580.3 (34.4–9780.9)4.7 × 10^−24^[62]European80 (34–187)<10^−6^[63]Japanese62.8 (21.2–185.8)5.388 × 10^−12^[67]Thai348.3 (19.2–6336.9)1.6 × 10^−13^[65]579.0 (29.5–11,362.7)<0.001[66]DRESSHLA-B*58:01Han Chinese47.7 (18.2–125.4)1.0 × 10^−26^[71]

Thai430.3 (22.6–8958.9)<0.001[66]SCARsHLA-B*58:01Korean97.8 (18.3–521.5)2.45 × 10^−11^[68]HLA-B*58:01Han Chinese44.0 (21.5–90.3)2.6 × 10^−41^[71]European39.11 (4.49–340.51)5.9 × 10^−4^[64]DILIHLA-B*58:01, HLA-B*53:01 clusterAfrican-American, Caucasian, HispanicNA0.0007[69]MPEHLA-B*58:01Thai144.0 (13.9–1497.0)<0.001[66]Han Chinese8.5 (4.2–17.5)2.3 × 10^−9^[71]Abbreviations: SJS/TEN, Stevens–Johnson syndrome/toxic epidermal necrolysis; DRESS, drug reaction with eosinophilia and systemic symptoms; DILI, drug-induced liver injury; SCARs, severe cutaneous adverse reactions; MPE, maculopapular exanthema; NA, not available; OR, odds ratio.

### 4.3. Genetic Susceptibility of Antibiotics-Induced Hypersensitivity Reactions

Antibiotics can cause either immediate-type or delayed-type drug hypersensitivity. Antibiotic hypersensitivity has shown an immune-related genetic predisposition. The HLA-DRB3*02:02 allele, absent in Europeans, accounts for 83% of amoxicillin-induced MPE cases in Italy [72] (Table 4). A high level of HLA-DRB1*15:01 was observed in Europeans with amoxicillin-clavulanate-induced liver injury [73], and a high level of HLA-B*57:01 was observed in flucloxacillin-induced liver injury [74].

Co-trimoxazole, a combination of sulfamethoxazole (SMX) and trimethoprim (TMP), is associated with delayed drug hypersensitivity. HLA-B*38 was reported to be related to sulfamethoxazole-induced SJS/TEN in Europeans [63]. Kongpan T. et al. reported that carriers with HLA-B*15:02, HLA-C*06:02, or HLA-C*08:01 had an increased risk of co-trimoxazole-induced SJS/TEN (odds ratio: 11) [75]. By whole genome sequencing (WGS), our recent multi-country case-control study showed that HLA-B*13:01 was strongly associated with co-trimoxazole-induced SCARs in patients from Taiwan, Thailand, and Malaysia [76]. Notably, HLA-B*13:01 contributed to 85.4% of patients with co-trimoxazole-induced DRESS [76]. A multicentric study of the Thai population showed that HLA-B*15:02 and HLA-C*08:01 are associated with cotrimoxazole-SJS/TEN and HLA-B*13:01 in DRESS. Additionally, the haplotypes of HLA-A*11:01-B*15:02 and HLA-B*13:01-C*03:04 are associated with co-trimoxazole-induced SJS/TEN and DRESS, respectively [77]. Earlier studies indicate that gene variants involved in drug metabolisms, such as NAT2 [78,79,80,81,82] and GSTM1 [83] null genotypes, were associated with sulfonamide-induced hypersensitivity reactions. However, the associations were weak and lacked validation.

Other significant discoveries, regarding pharmacogenomic associations with antibiotics-induced SCARs, include HLA-A*32:01, is associated with vancomycin-induced DRESS in Caucasians [84], and HLA-B*13:01, associated with dapsone-induced DRESS in north-eastern and south-eastern Asians [85,86,87,88,89,90] (Table 4).
biomedicines-11-00177-t004_Table 4Table 4Genetic variants associated with antibiotics-induced hypersensitivity reactions.Causative DrugsReactionsGenetic FactorsEthnicityOR(95% CI)*p*-ValueReferenceAmoxicillinMPEHLA-DRB3*02:02European8.88 (3.37–23.32) ^1^<0.0001[72]Co-amoxiclavDILIHLA-DRB1*15:01European2.59 (1.44–4.68)0.002[73]FlucloxacillinDILIHLA-B*57:01European80.6 (22.8–284.9)8.97 × 10^−19^[74]Sulfamethoxazole (SMX)SJS/TENHLA-B*38European8.6 (3.5–21)<0.003[63]Co-trimoxazole (SMX/TMP)SJS/TENHLA-B*15:02Thai3.91 (1.42–10.92)0.0037[75]HLA-C*06:0211.84 (1.24–566.04)0.0131HLA-C*08:013.53 (1.21–10.40)0.0108SCARsHLA-A*11:01-B*15:02 haplotype, Thai (HIV-infected patients only)4.36 (1.43–13.34)0.0108[77]HLA-B*13:01-C*03:04 haplotype3.77 (1.27–11.19)0.0251HLA-B*13:01Han Chinese, Thai, Malaysia11.7 (5.7–24)1.3 × 10^−13^[76]VancomycinDRESSHLA-A*32:01EuropeanNA1 × 10^−8^[84]DapsoneSCARsHLA-B*13:01Thai39.00 (7.67–198.21)5.34 × 10^−7^[85]54.00 (7.96–366.16)0.0001[88]DRESSHLA-B*13:01Han Chinese20.53 (11.55–36.48)6.84 × 10^−25^[86]Taiwanese, Malaysian49.64 (5.89–418.13)2.92 × 10^−4^[87]HLA-B*13:01Korean73.67 (2.56–2119.93)0.012[89]HLA-B*13:01Papua233.46 (1.7–67.7)7.11 × 10^−9^[90]Abbreviations: SJS/TEN, Stevens–Johnson syndrome/toxic epidermal necrolysis; DRESS, drug reaction with eosinophilia and systemic symptoms; SCARs, severe cutaneous adverse reactions; NA, not available; OR, odds ratio. ^1^ The odds ratio represents the increased risk of delayed type compared to immediate type reaction.

### 4.4. Genetic Susceptibility of Antiviral Agents-Induced Hypersensiticity

Abacavir is a nucleoside reverse transcriptase inhibitor, usually used in combined therapy for treating patients with HIV infection. Approximately 5–8% of European patients treated with abacavir developed immune-related ADR during the first six weeks of treatment [91]. HLA-B*57:01 was identified as a genetic predisposition for abacavir-related hypersensitivity in Caucasians in 2002 [92,93] (Table 5). The subsequent randomized clinical trials, recruiting 1956 patients from 19 countries, demonstrated that the carriage of HLA-B*57:01 could be a genetic predictor, in order to prevent abacavir hypersensitivity [94].

Nevirapine, a non-nucleoside reverse transcriptase inhibitor to treat HIV infection, has been reported to be associated with a hypersensitivity reaction; it has the clinical presentations of fever, a skin rash, or hepatitis. HLA-B*35:05 and HLA-Cw*04:01 were found to be related to a nevirapine-induced skin rash in Thailand [95,96] and nevirapine-induced SJS/TEN among Africans, respectively [97] (Table 5). Of note, HLA-Cw*04 was also found to be associated with cutaneous adverse reactions in multiple ethnicities [96] (Table 5). HLA-DRB1*01:01 was proposed to contribute to nevirapine-induced DRESS in patients in West Australia [98] (Table 5). Studies of multiple ethnicities show that the HLA-DRB1*01:01 allele is associated with DILI in white people [96] (Table 5). The other studies showed HLA-Cw*08 was associated with Nevirapine-induced hepatitis in Sardinian and Japanese populations [99,100]. The association between nevirapine-induced hepatotoxicity and HLA-Cw*04 was reported in Han Chinese people; this needs further validation [101]. In addition, an SNP (rs3099844) of the HCP gene was proposed to be associated with nevirapine-SJS/TEN in Africans [102] (Table 5).

Raltegravir, an HIV integrase inhibitor introduced in 2007, was associated with DRESS in Africans. HLA-B*53:01 was implicated as a risk allele of raltegravir-DRESS in the African population [103] (Table 5).
biomedicines-11-00177-t005_Table 5Table 5Genetic variants associated with antiviral agents-induced hypersensitivity reactions.Causative DrugsReactionsGenetic FactorsEthnicityOR(95% CI)*p*-ValueReferenceAbacavirDiHSHLA-B*57:01Caucasians23.6 (8.0–70.0)<0.0001[92]117 (29–481)<0.0001[93]NANA[94]NevirapineSJS/TENrs3099844 (HCP5)Mozambique2.03 (na)0.039[102]HLA-C*04:01African4.84 (2.71–8.61)8.47 × 10^−8^[97]DiHSHLA-Cw*04Thai2.43(1.22–4.84)0.17[96]NA0.0088[104]Asians, Blacks, Whites2.51 (1.73–3.62)8.7 × 10^−6^[96]Han Chinese3.611 (1.135–11.489)0.030[101]HLA-B*35:05Thai18.96 (4.87–73.44)4.6 × 10^−6^[95]HLA-B*35Asians3.47 (1.58–7.61)0.053[96]HLA-DRB1*01Whites3.02 (1.66–5.49)0.0074[96]4.8 (na)0.14[98]HLA-Cw*08SardinianNA0.05[99]JapaneseNA0.03[100]RaltegravirDRESSHLA-B*53:01AfricanNANA[103]Abbreviations: SJS/TEN, Stevens–Johnson syndrome/toxic epidermal necrolysis; DiHS, drug-induced hypersensitivity syndrome; DRESS, drug reaction with eosinophilia and systemic symptoms; NA, not available; OR, odds ratio.

### 4.5. Genetic Susceptibility of Hypersensitivity Reactions to Anti-Thyroid Drugs and Methazolamide

Anti-thyroid drugs (ATD), including carbimazole and methimazole, have been reported to induce agranulocytosis, and their association with HLA genotypes has been found in different ethnicities. Table 6 lists the genetic variants associated with drug-induced agranulocytosis (DIA). Methimazole-induced agranulocytosis was associated with HLA-DRB1*08:03:02 in Japanese people [105] (Table 6). HLA-B*27:05, HLA-B*38:02, and HLA-DRB1*08:03 alleles were found to be related to ATD-induced agranulocytosis in Taiwan [106] (Table 6). HLA-B*38:02 and HLA-DRB1*08:03 alleles were reported to be associated with ATD-induced agranulocytosis in Han Chinese people [107,108] (Table 6). HLA-B*27:05 was reported in a European population [109], as well as in Han Chinese people from northern China [110] (Table 6).

Methazolamide, an intraocular pressure-lowering drug, may cause SJS/TEN in Asians. HLA-B*59:01 has been proposed to be associated with methazolamide-induced SJS/TEN in Korean, Japanese, and Han Chinese patients [111,112,113,114] (Table 6).
biomedicines-11-00177-t006_Table 6Table 6Genetic variants associated with hypersensitivity reactions to anti-thyroid drugs and methazolamide.Causative DrugsReactionsGenetic FactorsEthnicityOR//break//(95% CI)*p*-ValueReferenceMethimazoleDIAHLA-DRB1*08:03:02Japanese5.42 (na)0.002[105]HLA-B*38:02, DRB1*08:03 haplotypeHan Chinese48.41 (21.66–108.22)3.32 × 10^−21^[107]HLA-B*38:0221.48 (11.13–41.48)6.75 × 10^−32^HLA-DRB1*08:036.13 (3.28–11.46)1.83 × 10^−9^Carbimazole/MethimazoleDIAHLA-B*38:02:01Han Chinese265.5 (27.87–2528.0)2.5 × 10^−14^[108]Carbimazole, Methimazole, PropylthiouracilDIAHLA-B*27:05Caucasian7.30 (3.81–13.96)1.91 × 10^−9^[109]Methimazole, PropylthiouracilDIAHLA-B*27:05Han Chinese60.11 (3.27–1104.4)1.1 × 10^−4^[110]HLA-B*38:026.55 (2.11–20.36)2.41 × 10^−4^HLA-DRB1*08:033.95 (1.60–9.79)1.57 × 10^−3^MethazolamideSJS/TENHLA-B*59:01Han Chinese305.0 (11.3–8259.9)6.3 × 10^−7^[111]Korean249.8 (13.4–4813.5)<0.001[112]JapaneseNANA[114]HLA-B*59:01Han Chinese146.00 (16.12–1321.98)6.19 × 10^−10^[113]HLA-B*55:0271.00 (7.84–643.10) 1.43 × 10^−4^Abbreviations: DIA, drug-induced agranulocytosis; SJS/TEN, Stevens–Johnson syndrome/toxic epidermal necrolysis; NA, not available; OR, odds ratio.

## 5. T Cell Receptor (TCR) Usage in Delayed Drug Hypersensitivity

In addition to HLA alleles, TCRs also play a crucial role in the pathogenesis of drug hypersensitivity. The preferential usage of TRBV genes and clonally-expanding CDR3 was observed in blister cells from skin lesions, and oxypurinol-cultured peripheral blood mononuclear cells of allopurinol-SCAR patients [115]. Recently, we identified a public αβ T cell receptor (TCR) from the skin blister cells of CBZ-SJS/TEN Asian and European patients [116]. The public TCR was composed of VFDNTDKLI and ASSLAGELF of CDR3 in TRA and TRB chains, respectively [116] (Table 7). This clonotype of TCR showed drug- and phenotype-specificity in an HLA-B*15:02-favored manner. Introducing T cells, with this TCR clonotype, to HLA-B*15:02 transgenic mice via the oral administration of CBZ, resulted in the development of SCARs symptoms. By comparison, HLA-B*15:02 transgenic mice received CBZ, but no adoptive T cell transfer showed SCAR symptoms. The data suggests that specific TCR recognizes the drug antigen and participates in SCAR. In addition, the results also support that HLA is insufficient to induce SCAR, and without the presence of drug-specific TCR, HLA-B*15:02 carriers are tolerant to CBZ [116]. Furthermore, a specific αβTCR pair was observed in HLA-B*13:01-restricted dapsone-induced drug-induced hypersensitivity syndrome (DiHS) [117]. Dapsone interacts with both HLA-B*13:01 and a specific TCR clonotype, the pair of TRAV12-3 and TRBV28 [117] (Table 7). The mode of interaction between dapsone, HLA, and TCR is different from that of abacavir and HLA-B*57:01, but similar to the interaction between oxypurinol and HLA-B*58:01.

## 6. Key Immune Mediators Involved in Delayed Drug Hypersensitivity

T lymphocytes-mediated delayed-type drug hypersensitivity reactions trigger the activation and production of many cytokines and chemokines, such as TNFs, IFNs, GM-CSF, TARC/CCL17, IL-6, IL-8/CXCL8, IL-4, IL-5, IL-8, IL-15, IL-36, RANTES, and CXCL8, etc. These cytokines/chemokines could enhance more cytotoxic cells, including macrophages, eosinophils, neutrophils, and mast cells, gathering and functioning in the inflammatory site and leading to tissue damage. These cytokines and chemokines are responsible for the trafficking, proliferation, regulation, or activation of T lymphocytes and other leukocytes. For example, IL4 and IL-5 play the main role in type IVb reactions by regulating the proliferation, migration, and activation of eosinophils [118]. Neutrophils are the main mediator cells in the type IVd reaction and could be activated and recruited by IL-8, CXCL8, GM-CSF, RANTES, MIP-2, and TNF-α [119,120,121,122,123].

For AGEP, the accumulation of neutrophils is the main characteristic, and CXCL8/IL-8 plays an essential role in the neutrophil-forming pustules [122]. IL-7, IL-22, and GM-CSF might synergistically enhance CXCL8/IL-8 production, and prevent neutrophil apoptosis [124]. Neutrophils, macrophages and mast cells could be identified from the skin lesions of patients with AGEP, implicating the involvement of different innate immune cells downstream of delayed drug hypersensitivity [124]. DILI was hypothesized to begin with damage-associated molecular pattern molecules (DAMPs), such as HMGB1 and ATP, and the activation of T cells [125]. DAMPs, activated by innate immune responses, can cause the activation of cytotoxic cells; this releases TNFs, IL-1b, IL-8, IL-6, and CXCL10, thus recruiting more leukocytes [126]. The histology of DILI suggests that the cellular immune response mainly involves T_H_1 and CD8+ T lymphocytes [127]. In addition, DILI frequently shows eosinophilia, suggesting the involvement of T_H_2 and IL-5 signaling [128]. These studies suggest that the complex interaction and cross-talk of innate and adaptive immunity are involved in the clinical presentation of delayed drug hypersensitivity [122,126].

Cytotoxic T lymphocytes (CTL) play a major role in type IVc hypersensitivity, and preferential TCR may recognize specific antigens represented by HLA molecules, leading to the direct killing of antigen-presenting cells. The cellular contact of target cells and effector cells (CTL) could induce the death of antigen-presenting cells through two proposed mechanisms, i.e., the delivery of cytotoxic proteins (e.g., granulysin, perforin, and granzyme B) and Fas-FasL signaling [129]. Perforins and granzymes are the major types of cytotoxic proteins released by CTL. The polymer of perforin forms pores in the cell surface of the target cell in the presence of Ca^2+^, and causes cell lysis and the cell membrane to become permeable for the entry of granzymes [130]. We previously identified a cytotoxic protein, granulysin, as a key mediator for keratinocyte death in SJS/TEN [131]. In addition, the interaction of Fas on the target cell membrane with the Fas ligand, expressed on the CTL cell surface, has been reported to induce caspase-dependent target cell apoptosis in TEN [5,132,133]. The key mediators of type IVc drug hypersensitivity are summarized below.

(1)Granulysin

Granulysin (GNLY), originally known as an anti-microbial peptide, is a member of the saposin-like protein (SAPLIP) family. It is also a component of lytic granules in CTL and nature killer (NK) cells. GNLY was demonstrated to be potently adept in lysing bacteria extracellularly, and to be more efficient with additional perforin and granzyme B in eliminating intracellular bacteria [134]. Opposite to granzyme B, which induces apoptosis via caspase-3 and -9, granulysin causes endoplasmic reticulum stress and activates caspase-7 [135]. We first reported that 15 kDa secretory granulysin serves as a key mediator for the disseminated keratinocyte apoptosis in patients with SJS/TEN [131]. Compared to the perforin, granzyme B, and FasL, the levels of granulysin were significantly increased in SJS/TEN blister fluids. The overexpression or depletion of granulysin was correlated to the cell cytotoxicity in the models of SJS/TEN [131]. Many studies support that granulysin is aggressively enhanced in drug-induced SJS/TEN, FDE, and DRESS/DiHS, but not MPE [136,137,138].

(2)Perforin and granzyme B

The activated drug-specific CTL and NK cells could release perforins to punch pores on the membrane of target cells, which promotes the entry of granzymes to activate the caspase cascade and induce apoptosis [139]. Granzymes are serine proteases with five types (A, B, H, K, and M) in humans, and they could induce cell death in different pathways. Granzyme A and granzyme B are abundantly expressed in CTL and NK cells. They penetrate into target cells through perforin pores and cause cell death in the classical caspase apoptotic pathway by granzyme B or in a caspase-independent pathway by granzyme A [140]. Granzyme B is the greatest pro-apoptotic member in the granzyme family and mainly contributes to DNA fragmentation in the susceptible cell. Granzyme B mediates caspase-dependent cell death by directly activating pro-caspases and cleaving downstream caspase substrates, such as an inhibitor of caspase-activated DNase (ICAD). Despite direct activation of procaspase-3, granzyme B specifically and rapidly cleaves Bid into a truncated form and induces the release of cytochrome c and Smac/Diablo to activate the caspase-3 pathway [141]. The other caspase-independent cell death mechanism implies that granzyme B causes cell death with an abolished caspase activity [142].

(3)Fas/FasL signaling pathway

Fas ligand (FasL), a member of the TNF family, induces apoptosis in susceptible cells in response to the cross-linking of the receptor, Fas. Fas/FasL-induced apoptosis plays an essential role in immune homeostasis and is involved in cytotoxicity in epidermal cells in drug hypersensitivity. The Fas-associated death domain protein (FADD) is recruited to Fas upon the interaction of Fas and FasL. The binding of procaspase-8 to FADD results in the formation of the death-inducing signaling complex, finally leading to the activation of effector caspase-3 through activated caspase-8 [143]. Viard et al. proposed that a suicidal interaction between Fas and FasL, expressed in keratinocytes, resulted in the extensive necrosis of epidermal cells in patients with SJS/TEN [144].

(4)Thymus and activation-regulated chemokine (TARC) and type 2 helper T cells (T_H_2)

TARC is expressed by monocyte-derived dendritic cells [145] and epithelial cells [146]. It can regulate the migration and activation of the type 2 T helper (T_H_2) via CCR4 [147]. The serum level of TARC was found to be significantly associated with blood eosinophil counts and the severity of DiHS/DRESS [148,149]. TARC has been suggested to be a prognostic marker for early DiHS/DRESS [150]. Additionally, the finding of eosinophilia, the increase in T_H_-2-associated cytokines and chemokines (e.g., TARC and macrophage-derived chemokine (MDC)), and the high proportion of IL-4 and IL-13-producing CD4+ T cells in DiHs/DRESS, suggest that T_H_2 cells play an essential role in the pathogenesis of DiHS/DRESS [148,151,152,153,154].”

(5)Regulatory T cell (Treg)

The regulatory T cell (Treg) has been suggested to be involved in the pathogenesis of delayed drug hypersensitivity. Takahashi et al. found that the frequency of Tregs in skin lesions was not changed, but the function was impaired in toxic epidermal necrolysis (TEN) patients. Opposite to the observation of TEN, functional Treg dramatically expanded and was abundantly located in skin lesions of DiHS/DRESS patients. The number of Tregs decreases, and Tregs becomes functionally impaired upon the resolution of DiHS/DRESS [155]. Different cytokine expressions in the microenvironment may cause the contraction of Tregs. For example, IL-6, released from CD16+ monocytes, can turn Tregs into T_H_17 [156]. Such a shift may explain the development of an autoimmune response in a prolonged period of DiHS/DRESS, after clinical resolution [156].

## 7. Conclusions

Delayed-type hypersensitivity reactions are mainly mediated by the T cell recognition of drug antigens and are accompanied by the activation of downstream leukocytes and immune mediators. These immune responses lead to the diverse clinical presentations of delayed drug hypersensitivity reactions, which range from mild skin reactions (e.g., MPE and FDE) to life-threatening ADRs (e.g., SJS, TEN, DRESS, DiHS, DILI, and AGEP). Several medicines, including anti-epileptics, antibiotics, anti-gout, and anti-viral agents, are associated with delayed drug hypersensitivity reactions. The formation of an immune synapse, composed of TCR, drug/metabolite/peptide, and HLA, may trigger the molecular recognition of delayed drug hypersensitivity. There are four proposed models for the molecular recognition of delayed drug hypersensitivity: the hapten/pro-hapten hypothesis, the pharmacological interaction with immune receptor (p-i) concept, the altered peptide repertoire model, and the altered TCR repertoire model. In these models, the causative drug/metabolite may interact covalently or noncovalently with peptides/protein, and bind to HLA and/or TCR to elicit drug hypersensitivity reactions. Increasing pharmacogenomic studies reveal that genetic variants of HLA loci and drug metabolic enzymes are associated with delayed drug hypersensitivity. Furthermore, preferential TCR clonotypes, and cytotoxic proteins/cytokines/chemokines’ activation, have been reported to contribute to the pathogenesis of delayed drug hypersensitivity. The recent findings, regarding specific T cell receptors in allopurinol-, carbamazepine-, and dapsone-induced SCARs, support the attribution of drug-specific T cells in delayed drug hypersensitivity. Further studies on molecular recognition, genetic susceptibility, and immune mediators provide an important knowledge basis for preventing, diagnosing, and the clinical managing of delayed-type drug hypersensitivity.

## Figures and Tables

**Figure 1 biomedicines-11-00177-f001:**
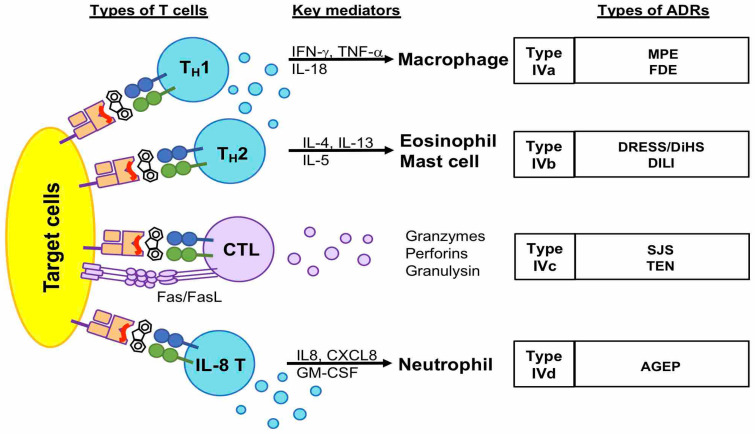
The characteristics of type IV (delayed-type) drug hypersensitivity. Delayed-type drug hypersensitivity is mostly induced by T cells. They are subdivided into four types (IVa, IVb, IVd, and IVc) according to the types of T cells, such as T_H_1, T_H_2, IL8 T_H_ cells, and cytotoxic T cells. The types IVa, IVb, and IVd are mediated by T_H_ cells and the activation of downstream granulocytes, such as macrophages, mast cells, eosinophils, and neutrophils. By comparison, type IVc is mainly mediated by cytotoxic T cells, which induce target cell death by releasing cytokines, such as granulysin, granzyme B, and perforin, or direct interaction of Fas/FasL. Abbreviations: FDE, fixed drug eruption; MPE, maculopapular eruption; AGEP, acute generalized exanthem pustulosis; DRESS, drug reactions with eosinophilia and systemic symptoms; DiHS, drug-induced hypersensitivity syndrome; DILI, drug-induced liver injury; SJS, Stevens–Johnson syndrome; TEN, toxic epidermal necrolysis.

**Figure 2 biomedicines-11-00177-f002:**
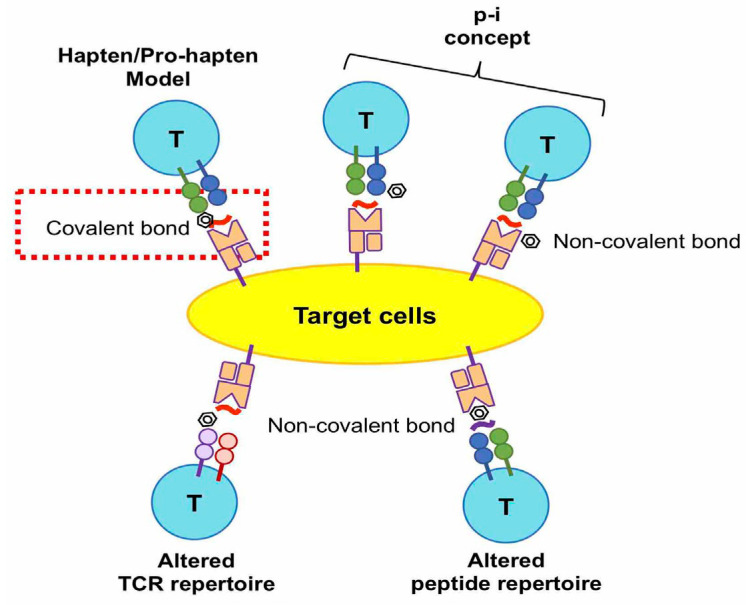
Four hypotheses proposed for the molecular recognition of drugs by TCR in delayed type drug hypersensitivity. The hapten model hypothesizes that a drug or metabolite covalently binds to an endogenous peptide and leads to an immunogenic response. The other three hypotheses suggest non-covalent interactions between drugs, endogenous peptides, TCR, and HLA. The pharmacological interaction with the immune receptor (p-i) concept hypothesizes that the drug interacts with TCR, HLA, or both non-covalently. Altered TCR repertoire and altered peptide repertoire models propose that the drug non-covalently binds on TCR or HLA in the groove region.

**Table 1 biomedicines-11-00177-t001:** Classification and clinical symptoms of drug hypersensitivity reactions based on the definition by ICON.

ICON Classification	Reactions	Causality
Immediate type	Angioedema, Urticaria, Anaphylaxis	NSAIDs, NMBA, Aspirin, Antibiotics, Vaccines, etc.
Non-immediate type	MPE, FDE, DRESS/DiHS, SJS/TEN, AGEP, DILI	AEDs, Antibiotics, Anti-viral agents, Allopurinol, NSAIDs, etc.

Abbreviations: AEDs, antiepileptic drugs; AGEP, acute generalized exanthematous pustulosis; DILI, drug-induced liver injury; DiHS, drug-induced hypersensitivity syndrome; DRESS, drug reaction with eosinophilia and systemic symptoms; FDE, Fixed drug eruption; MPE, Maculopapular exanthema; NMBAs, neuromuscular blocking agents; NSAIDs, nonsteroid anti-inflammation drugs; SJS/TEN, Stevens–Johnson syndrome/toxic epidermal necrolysis.

**Table 7 biomedicines-11-00177-t007:** TCR clonotype usage in SCARs.

HLA Types	Drug	TCR Clonotype	Reactions	Reference
HLA-B*15:02	Carbamazepine	TRBV12-4-TRBJ2-2, ASSLAGELF/ TRAV12-1-TRAJ34, VFDNTDKLI	SJS/TEN	[116]
HLA-B*13:01	Dapsone	TRAV12-3/TRBV28 pair	SJS, DiHS	[117]

Abbreviations: SJS/TEN, Stevens–Johnson syndrome/toxic epidermal necrolysis; DiHS, drug-induced hypersensitivity syndrome.

## Data Availability

Not applicable.

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
