# Peer review of "Delayed Drug Hypersensitivity Reactions: Molecular Recognition, Genetic Susceptibility, and Immune Mediators"

_biomedicines, 2023, doi:10.3390/biomedicines11010177_

Round 1
Reviewer 1 Report
In this manuscript, Chu et al. provide a narrative review on the molecular basis, genetic susceptibility, and immune mediators involved in delayed drug hypersensitivity reactions to select anti-epileptic drugs, allopurinol, antibiotics, antiviral drugs, anti-thyroid drugs, and methazolamide. This review seems to be outside the scope of Biomedicines, which focuses on such topics as "pathogenesis mechanisms of diseases, translational medical research, biomaterial in biomedical research, natural bioactive molecules, biologics, biosimilar, vaccines, gene therapies, cell-based therapies, targeted specific antibodies, recombinant therapeutic proteins, nanobiotechnology driven products, targeted therapy, bioimaging, biosensors, biomarkers, biosimilars, and nano-biosimilars." This review would probably receive more readership interest in another journal with more of a focus on immunology/hypersensitivities. My general comments are listed below.
(1) Line 121 under Section 3: This sentence is unclear as written. The authors are talking about the formation of a covalent bond between the beta-lactam ring of a beta-lactam antibiotic and an endogenous protein. Please restructure for clarity.
(2) Table 3: Delete the "Causative drugs" column since you are only presenting information on allopurinol-induced hypersensitivity reactions.
(3) Table 4: Explain what superscript 1 refers to in "DRESS1".
(4) Table 6: Explain what superscript 1 refers to in "DIA1".
Author Response
Point-to-point response letter:
Reviewer 1:
Comments and Suggestions for Authors
In this manuscript, Chu et al. provide a narrative review on the molecular basis, genetic susceptibility, and immune mediators involved in delayed drug hypersensitivity reactions to select anti-epileptic drugs, allopurinol, antibiotics, antiviral drugs, anti-thyroid drugs, and methazolamide. This review seems to be outside the scope of Biomedicines, which focuses on such topics as "pathogenesis mechanisms of diseases, translational medical research, biomaterial in biomedical research, natural bioactive molecules, biologics, biosimilar, vaccines, gene therapies, cell-based therapies, targeted specific antibodies, recombinant therapeutic proteins, nanobiotechnology driven products, targeted therapy, bioimaging, biosensors, biomarkers, biosimilars, and nano-biosimilars." This review would probably receive more readership interest in another journal with more of a focus on immunology/hypersensitivities. My general comments are listed below.
(1) Line 121 under Section 3: This sentence is unclear as written. The authors are talking about the formation of a covalent bond between the beta-lactam ring of a beta-lactam antibiotic and an endogenous protein. Please restructure for clarity.
Reply: Thanks for the comment. We have revised the sentence.
“The major determinant of penicillin-induced hypersensitivity is penicilloyl polylysine. This structure is formed by covalent bound of b-lactam ring to lysine residues in proteins.”
(2) Table 3: Delete the "Causative drugs" column since you are only presenting information on allopurinol-induced hypersensitivity reactions.
Reply: The “causative drugs” column is deleted.
(3) Table 4: Explain what superscript 1 refers to in "DRESS1".
Reply: The superscript 1 is a redundant character and is deleted. It was labeled to describe the abbreviation of DRESS.
(4) Table 6: Explain what superscript 1 refers to in "DIA1".
Reply: The superscript 1 is a redundant character and is deleted. It was labeled to describe the abbreviation of DIA.
Reviewer 2 Report
This paper is well written, and the text is clear and easy to read. I have some comments.
1. At the beginning of the abstract, the authors may consider a different starting word for the first three consecutive "Drug hypersensitivity reactions".
2. Please clarify the odds ratios in Tables 2-7.
3. The authors may add the relationship between TARC and drug hypersensitivity reactions, as in the following papers.
Ogawa K, Morito H, Hasegawa A, Daikoku N, Miyagawa F, Okazaki A, Fukumoto T, Kobayashi N, Kasai T, Watanabe H, Sueki H, Iijima M, Tohyama M, Hashimoto K, Asada H. Identification of thymus and activation-regulated chemokine (TARC/CCL17) as a potential marker for early indication of disease and prediction of disease activity in drug-induced hypersensitivity syndrome (DIHS)/drug rash with eosinophilia and systemic symptoms (DRESS). J Dermatol Sci. 2013 Jan;69(1):38-43.
Ogawa K, Morito H, Hasegawa A, Miyagawa F, Kobayashi N, Watanabe H, Sueki H, Tohyama M, Hashimoto K, Kano Y, Shiohara T, Ito K, Fujita H, Aihara M, Asada H. Elevated serum thymus and activation-regulated chemokine (TARC/CCL17) relates to reactivation of human herpesvirus 6 in drug reaction with eosinophilia and systemic symptoms (DRESS)/drug-induced hypersensitivity syndrome (DIHS). Br J Dermatol. 2014 Aug;171(2):425-7.
Komatsu-Fujii T, Kaneko S, Chinuki Y, Suyama Y, Ohta M, Niihara H, Morita E. Serum TARC levels are strongly correlated with blood eosinophil count in patients with drug eruptions. Allergol Int. 2017 Jan;66(1):116-122.
Komatsu-Fujii T, Chinuki Y, Niihara H, Hayashida K, Ohta M, Okazaki R, Kaneko S, Morita E. The thymus and activation-regulated chemokine (TARC) level in serum at an early stage of a drug eruption is a prognostic biomarker of severity of systemic inflammation. Allergol Int. 2018 Jan;67(1):90-95.
4. The regulatory T cells in drug hypersensitivity reactions would be mentioned.
5. If the authors don't mind, how about adding the epidemiology and treatment for drug hypersensitivity reactions? This information is useful for readers.
Author Response
Reviewer 2:
Comments and Suggestions for Authors
This paper is well written, and the text is clear and easy to read. I have some comments.
- At the beginning of the abstract, the authors may consider a different starting word for the first three consecutive "Drug hypersensitivity reactions".
Reply: Thanks for the comment. We have revised the sentences.
“Drug hypersensitivity reactions are classified into immediate and delayed types according to the onset time. In contrast to the immediate type, delayed drug hypersensitivity mainly involves T lymphocyte recognition of the drug antigens and cell activation. The clinical presentations of such hypersensitivity are various and range from mild reactions (e.g., maculopapular exanthema (MPE) and fixed drug eruption (FDE)) to drug-induced liver injury (DILI) and severe cutaneous adverse reactions (SCARs) (e.g., Stevens-Johnson syndrome (SJS), toxic epidermal necrolysis (TEN), drug reaction with eosinophilia and systemic symptoms (DRESS), and acute generalized exanthematous pustulosis (AGEP))..
- Please clarify the odds ratios in Tables 2-7.
Reply: We have added new columns describing odds ratio (OR) (95% CI) and P-values in Tables 2-6. There is no OR and P-value reported in the original studies in Table 7.
- The authors may add the relationship between TARC and drug hypersensitivity reactions, as in the following papers.
Ogawa K, Morito H, Hasegawa A, Daikoku N, Miyagawa F, Okazaki A, Fukumoto T, Kobayashi N, Kasai T, Watanabe H, Sueki H, Iijima M, Tohyama M, Hashimoto K, Asada H. Identification of thymus and activation-regulated chemokine (TARC/CCL17) as a potential marker for early indication of disease and prediction of disease activity in drug-induced hypersensitivity syndrome (DIHS)/drug rash with eosinophilia and systemic symptoms (DRESS). J Dermatol Sci. 2013 Jan;69(1):38-43.
Ogawa K, Morito H, Hasegawa A, Miyagawa F, Kobayashi N, Watanabe H, Sueki H, Tohyama M, Hashimoto K, Kano Y, Shiohara T, Ito K, Fujita H, Aihara M, Asada H. Elevated serum thymus and activation-regulated chemokine (TARC/CCL17) relates to reactivation of human herpesvirus 6 in drug reaction with eosinophilia and systemic symptoms (DRESS)/drug-induced hypersensitivity syndrome (DIHS). Br J Dermatol. 2014 Aug;171(2):425-7.
Komatsu-Fujii T, Kaneko S, Chinuki Y, Suyama Y, Ohta M, Niihara H, Morita E. Serum TARC levels are strongly correlated with blood eosinophil count in patients with drug eruptions. Allergol Int. 2017 Jan;66(1):116-122.
Komatsu-Fujii T, Chinuki Y, Niihara H, Hayashida K, Ohta M, Okazaki R, Kaneko S, Morita E. The thymus and activation-regulated chemokine (TARC) level in serum at an early stage of a drug eruption is a prognostic biomarker of severity of systemic inflammation. Allergol Int. 2018 Jan;67(1):90-95.
Reply: Thanks for the comment. We have added a paragraph to discuss TARC and delayed drug hypersensitivity.
“(4) Thymus and activation-regulated chemokine (TARC) and type 2 helper T cells (Th2)
TARC is expressed by monocyte-derived dendritic cells [145] and epithelial cells [146]. It can regulate the migration and activation of type 2 T helper (Th2) via CCR4 [147]. The serum level of TARC was found to be significantly associated with blood eosinophil counts and the severity of DiHS/DRESS [148] [149]. TARC has been suggested to be a prognostic marker for early DiHS/DRESS [150]. Additionally, the findings of eosinophilia, increase of Th-2 associated cytokines and chemokines (e.g., TARC and macrophage-derived chemokine (MDC)), and the high proportion of IL-4 and IL-13-producing CD4+ T cells in DiHs/DRESS, suggest Th2 cells play an essential role in the pathogenesis of DiHS/DRESS [151][152] [153][154].”
4.The regulatory T cells in drug hypersensitivity reactions would be mentioned.
Reply: Thanks for the comment. We have added a paragraph to discuss Treg and delayed drug hypersensitivity.
(5) Regulatory T cell (Treg)
Regulatory T cell (Treg) has been suggested to be involved in the pathogenesis of delayed drug hypersensitivity. Takahashi et al.found that the frequency of Treg in skin lesions was not changed, but the function was impaired in toxic epidermal necrolysis (TEN) patients. Opposite to the observation on TEN, functional Treg dramatically expanded and abundantly located in skin lesions of DiHS/DRESS patients. The number of Treg decreases, and Treg becomes functionally impaired upon resolution of DiHS/DRESS [155]. Different cytokine expressions in the microenvironment may cause the contraction of Treg. For example, IL-6, released from CD16+ monocytes, can turn Treg into Th17 [156]. Such a shift may explain the development of autoimmune response in a prolonged period of DiHS/DRESS after clinical resolution [156].
- If the authors don't mind, how about adding the epidemiology and treatment for drug hypersensitivity reactions? This information is useful for readers.
Reply: Thanks for the comments. The epidemiology and treatment for drug hypersensitivity reactions have been discussed in other papers of this special issue.
- Chang HC et al. A Review of the Systemic Treatment of Stevens-Johnson Syndrome and Toxic Epidermal Necrolysis. Biomedicines. 2022 Aug 28;10(9):2105..
- Stirton H. et al. Drug Reaction with Eosinophilia and Systemic Symptoms (DReSS)/Drug-Induced Hypersensitivity Syndrome (DiHS)—Readdressing the DReSS. Biomedicines, 2022 Apr 26;10(5):999.
Round 2
Reviewer 1 Report
The authors did not respond to the following original comment by the reviewer: "This review seems to be outside the scope of Biomedicines, which focuses on such topics as 'pathogenesis mechanisms of diseases, translational medical research, biomaterial in biomedical research, natural bioactive molecules, biologics, biosimilar, vaccines, gene therapies, cell-based therapies, targeted specific antibodies, recombinant therapeutic proteins, nanobiotechnology driven products, targeted therapy, bioimaging, biosensors, biomarkers, biosimilars, and nano-biosimilars.' This review would probably receive more readership interest in another journal with more of a focus on immunology/hypersensitivities."